# The Impact of Integrated Reporting on the Cost of Capital: Evidence from an Emerging Market

Burak Pirgaip * and Lamija Rizvić

Department of Business Administration, Hacettepe University, Beytepe, Çankaya, Ankara 06800, Turkey; lamija393@gmail.com
* Correspondence: burakpirgaip@hacettepe.edu.tr

**Abstract:** The aim of this study is to investigate the influence of integrated reporting (IR) on the cost of financing within the Turkish capital market. Specifically, we analyze the effects of IR on the weighted average cost of capital (WACC), cost of equity (COE), and cost of debt (COD) for companies listed on Borsa Istanbul. Additionally, we explore how IR moderates the relationship between environmental, social, and governance (ESG) scores and the cost of financing. Our panel data analysis reveals a positive association between IR and both WACC and COD, while the impact on COE is not statistically significant. However, the findings suggest that the utilization of IR by companies to enhance the communication of their value-creating activities can mitigate WACC and COD, thus indicating a moderating effect on the relationship between ESG factors and the cost of financing.

**Keywords:** integrated reporting; ESG; cost of capital; Borsa Istanbul

## 1. Introduction

In recent years, growing concerns about climate change and environmental issues have led to increased public interest in corporate disclosure of non-financial information related to sustainability policies and practices. Recognizing the importance of such disclosures in fostering trust between companies and their stakeholders, new reporting frameworks have been developed to produce "sustainability reports" that incorporate environmental, social, and governance (ESG) data. However, the insufficient dissemination of information, especially on risks and uncertainties (Cabedo and Tirado 2004), through these reports has highlighted the need for a more comprehensive reporting system that promotes integrated thinking by integrating financial and non-financial information. Consequently, integrated reporting (IR) has emerged as an extremely important issue to improve the quality of information needed to create long-term value through an integrated thinking approach.

The main objective of IR is to enhance accountability in the utilization of six distinct forms of capital: financial, manufactured, intellectual, human, social and relationship, and natural capital (Busco et al. 2013). These different forms of capital have a direct impact on short-, medium-, and long-term positive or negative outcomes throughout the value chain encompassing business activities and outputs. As a result, IR plays a crucial role in a company's business model and governance, as these aspects are closely interconnected with its risks and opportunities, performance, strategy and resource allocation, and future prospects. Disclosing these fundamental components of value creation proves advantageous in facilitating decision-making by users of IR, enabling them to gain a comprehensive understanding of the company's overall societal impact.

The practice of IR is gaining traction around the world and attracting the attention of policymakers and academics. After South Africa introduced a mandatory regime based on the "apply or explain" principle in 2011, many countries are now voluntarily adopting IR, supported by professional institutions and major international companies. Scholars are

increasingly exploring various aspects of IR to demonstrate its contribution to the corporate landscape. While the existing literature on IR is abundant, it focuses predominantly on normative perspectives that discuss the merits and limitations of the practice, as outlined by Dumay et al. (2016). Vitolla et al. (2019) further support this argument, highlighting a recent shift towards assessing the determinants and effects of IR. Regarding its effects, extensive research has investigated the relationship between IR and firm performance, revealing positive impacts on stock market reactions (e.g., Lee and Yeo 2016; Nakajima and Inaba 2021), stock liquidity (e.g., Barth et al. 2017; Caglio et al. 2020), firm valuation (e.g., Baboukardos and Rimmel 2016; Tlili et al. 2019), and financial performance (e.g., Churet and Eccles 2014; Conway 2019).

One important strand of literature focuses on the relationship between IR and the cost of capital. This area of research has its roots in the existing work on the impact of reducing information asymmetry between companies and their investors and creditors on the cost of capital (Verrecchia 1983; Healy and Palepu 1993; Mazumdar and Sengupta 2005). Considering that companies are urged to provide comprehensive and informative disclosures to eliminate costly information asymmetries, mitigate agency costs, promote transparency, and prevent adverse selection, IR can serve as a valuable tool for exchanging information with stakeholders, enhancing transparency, and eliminating information asymmetry. By providing investors and creditors with additional information regarding value creation, IR can act as a signal that can result in a lower cost of capital.

Zhou et al. (2017) were among the first researchers to examine the influence of IR on the cost of capital. Their findings indicate that companies with high-quality IR tend to have lower costs of capital. The study focuses on a sample of 443 firm-year observations from 2009 to 2012 in the South African market, where IR is mandatory. García-Sánchez and Ligia (2017) discover a negative relationship between IR and the cost of capital in their study of 995 firms from 27 different countries over the period 2009–2013. Their findings suggest that the adoption of IR leads to a reduction in the cost of capital. The researchers also propose that IR, by addressing asymmetric information and enhancing transparency, can lower both the current and future costs of capital. Vena et al. (2020) conclude in their study that companies publishing IR can achieve a reduction in their cost of capital by approximately 1.4%. Notably, the impact of IR on the cost of capital is found to be more pronounced in countries characterized by high collectivism, low power distance, and high masculinity. The study includes samples from 31 different countries, comprising a total of 211 companies of various sizes and growth capabilities over the period 2009–2017. Maama and Marimuthu (2021) confirm the negative relationship between IR and the cost of capital based on their analysis of 148 listed companies across 10 sub-Saharan countries from 2009 to 2018. Their panel data analysis results align with the signaling theory, suggesting that companies signal positive information to the market when they provide details about their value creation, resulting in a reduction in the cost of capital.

Similar to the findings on the cost of capital, Vitolla et al. (2020) discover a connection between IR and the cost of equity (COE). Their study, which involves a sample of 116 international companies from Africa, the Americas, Asia, Europe, and Oceania demonstrated that the quality of IR can lower the COE. The authors propose that publishing IR conveys to investors that a company is not solely focused on financial performance, but also exhibits social and environmental responsibility, thereby attracting more long-term investors and potentially reducing the COE. Salvi et al. (2020) analyze a sample of 82 listed companies from 12 countries across Africa, Asia, Europe, and Oceania, and their findings support the notion that adequate representation of intellectual capital in IR allows companies to reduce their COE. The authors attribute this outcome to reduced information asymmetry, which enables investors to make more accurate decisions, thereby fostering greater confidence and lower COE.

Regarding the relationship between IR and the cost of debt (COD), Gerwanski (2020) finds, using a sample of 834 European companies from 2015 to 2017, that IR can lower a company's COD. The study also highlights that this effect is more significant for companies

with lower ESG performance, and is particularly relevant for companies operating in environmentally sensitive industries. Muttakin et al. (2020), in their study of South African companies from 2009 to 2015, found that companies implementing IR have lower COD compared to those that do not. The authors attribute this finding to the benefits of IR in reducing information gathering costs and monitoring costs. Raimo et al. (2022) employ a manual content analysis to evaluate the information quality of IR and a panel regression model to determine its impact on COD using a sample of 133 EU companies from 2017 to 2019. Their analysis reveals a negative relationship between the quality of IR and COD, suggesting that companies publishing high-quality IR can enjoy lower debt financing costs.

While these studies shed light on the IR-cost of financing relationship, the literature still remains relatively limited and emerging markets require particular attention. To address this gap, our study examines the impact of IR on the cost of capital for Turkish listed companies from 2015 to 2020. We not only assess the direct effect of IR but also explore its moderating influence on the link between ESG scores and the cost of capital. The characteristics of IR can influence the relationship between ESG scores and the cost of capital. IR, by integrating financial and non-financial information, including ESG disclosures, enhances the understanding of the link between ESG factors and the cost of capital. This moderating effect has already been recognized in previous studies (Atan et al. 2018; Wong et al. 2021; Nazir et al. 2021; Maama and Marimuthu 2021; Ramirez et al. 2022). Thus, it is plausible to assert that IR can contribute to making this relationship more evident and pronounced.

Our panel data analysis reveals an intriguing positive association between IR and the cost of capital. Additionally, our findings suggest that when sustainable companies utilize IR to effectively communicate their value-creating activities, the cost of capital can be reduced. Furthermore, while IR does not directly impact the COE, our study provides strong evidence of its significant relationship with COD.

Our study contributes in three key aspects. Firstly, we address the role of IR in the context of an emerging market, thus expanding the limited body of research on the link between IR and the cost of capital. Turkey has made significant strides in adopting IR since the first IR was published in 2015. Presently, there is growing interest and awareness of IR in Turkish capital markets, supported by institutional backing from the government and the business environment (see Appendix A). Secondly, we enhance the understanding of the cost of capital by examining its components, namely COE and COD. By analyzing these distinct elements, we aim to gain a deeper understanding of the intricate nature of the impact of IR and to assess whether it affects these costs in a similar manner. Thirdly, our analytical framework incorporates the possibility that IR may play a moderating role in the relationship between sustainability and the cost of capital. This aspect provides valuable insights into the significance of IR in fulfilling corporate social responsibility objectives and sheds light on how it influences the link between sustainability considerations and the cost of capital. By examining this moderating effect, we contribute to a comprehensive understanding of the role and importance of IR in promoting corporate sustainability practices.

The remainder of the paper is organized as follows. Section 2 describes the data and methodology. Section 3 presents the results. Section 4 discusses our findings. The conclusion is in Section 5.

## 2. Materials and Methods

Our data comprise a sample of 59 companies that are part of the BIST Sustainability Index (XUSRD), which comprises a list of companies with high sustainability performance. We focus specifically on these "sustainable" companies since they are the only ones that have IR preparers among them. We show the IR preparation status of these companies along with their industry information in Table 1.

**Table 1.** Sample companies and their IR preparing status.

| Industry | # of Companies | IR Preparers (# of Firm-Year obs.) | | | |
|---|---|---|---|---|---|
| | | **2017** | **2018** | **2019** | **2020** |
| Manufacturing | 22 | | | 1 | 5 |
| Financial | 20 | 1 | 3 | 5 | 7 |
| Energy | 5 | | | | |
| Wholesale-Retail | 4 | | | | |
| Technology | 3 | | | | |
| Transportation | 2 | | | | |
| Telecommunication | 2 | | | | 1 |
| Construction | 1 | | | | |
| Total | 59 | 1 | 3 | 6 | 13 |

Note: This table shows the industry information and IR preparation status of our sample companies.

Table 1 displays the distribution of the sample companies across various industries, with the majority operating in manufacturing (37.29%) and the financial sector (33.90%). Other sectors represented include energy (8.47%), wholesale-retail (6.78%), technology (5.08%), transportation (3.39%), telecommunications (3.39%), and construction (1.69%). Additionally, Table 1 presents the year-over-year increase in the number of firms adopting IR.

The sample period spans from 2015 to 2020, with 2015 as the starting year due to the unavailability of ESG scores and cost of capital data prior to that. Although some companies published new IR in 2021, these reports were excluded from our analysis as their corresponding ESG scores for that year were not accessible.

Table 2 provides a description of the variables of interest, outlined as follows:

**Table 2.** Variable description and sources.

| Variable | Symbol | Source |
|---|---|---|
| Dependent | | |
| Weighted average cost of capital (%) | WACC | Refinitiv |
| Cost of equity (%) | COE | Refinitiv |
| Cost of debt (%) | COD | Refinitiv |
| Independent | | |
| ESG score (grade) | ESG | Refinitiv |
| Environment pillar score (grade) | ENV | Refinitiv |
| Social pillar score (grade) | SOC | Refinitiv |
| Governance pillar score (grade) | GOV | Refinitiv |
| Integrated report (1 if an IR is published; 0 otherwise) | IR | ERTA |
| Total assets (TRY) | TA | Refinitiv |
| Total debt (ratio) | LEV | Refinitiv |
| Price-to-book ratio (ratio) | PB | Refinitiv |
| CDS (basis points) | CDS | Refinitiv |

Note: This table describes the variables of interest and the sources we obtain the data from.

The data for the sample companies is primarily sourced from the Refinitiv database, while information on IR is obtained from the IR Turkey Network (ERTA) website. Our study focuses on three dependent variables: WACC, COE, and COD. The key independent variables of interest are ESG scores, including scores for relevant pillars, and the presence of IR. The IR variable is a dummy variable that takes a value of 1 if the company published an IR in a given year, and 0 otherwise. We also include three firm-specific control variables and one macroeconomic control variable. The firm-specific control variables are total assets, total debt, and the price-to-book ratio, which have been used in prior studies (Bernardi and Stark 2018; Aouadi and Marsat 2018; Carp et al. 2019; Melegy and Alain 2020). Total assets reflect the size of the company, total debt indicates the level of leverage and is scaled by total assets, and the price-to-book ratio captures growth opportunities. As for

the macroeconomic control variable, we utilize Turkey's 5-year average credit default swap (CDS) premiums. All the data used in the analysis are collected on an annual basis. Descriptive statistics for the variables are presented in Table 3.

**Table 3.** Descriptive statistics.

| Variable | Obs. | Mean | Median | Std. Dev. | Min. | Max. |
|---|---|---|---|---|---|---|
| WACC | 223 | 0.087 | 0.086 | 0.023 | 0.035 | 0.154 |
| COE | 223 | 0.139 | 0.138 | 0.029 | 0.078 | 0.210 |
| COD | 223 | 0.032 | 0.030 | 0.014 | 0.006 | 0.074 |
| ESG | 223 | 59.331 | 61.750 | 15.237 | 17.380 | 93.950 |
| ENV | 223 | 59.399 | 61.790 | 21.313 | 0.000 | 97.440 |
| SOC | 223 | 65.247 | 67.840 | 19.588 | 14.070 | 97.320 |
| GOV | 223 | 52.402 | 51.370 | 19.635 | 11.500 | 94.370 |
| TA (billions TRY) | 223 | 13.313 | 2.654 | 26.394 | 0.243 | 118.737 |
| LEV | 223 | 0.313 | 0.307 | 0.175 | 0.000 | 0.903 |
| PB | 221 | 2.982 | 1.210 | 16.631 | 0.230 | 245.400 |
| CDS | 223 | 375.006 | 341.337 | 108.875 | 219.797 | 526.026 |

Note: This table summarizes the descriptive statistics of our data.

Table 3 reveals that WACC is predominantly influenced by COE rather than COD. COE exhibits the highest average value and volatility compared to other funding costs. The mean ESG/pillar scores indicate that the sample firms perform relatively well in meeting the social requirements of the ESG framework, although their performance in the governance pillar is comparatively lower. On average, the sample companies possess assets valued at approximately TRY 13 billion, with around 31% of their financing derived from debt. The price-to-book ratio suggests that the share price generally trades at a premium. To address potential heteroscedasticity concerns and normalize the data, we take the natural logarithm of the ESG/pillar scores, total assets, and CDS values.

In this study, we propose the following hypotheses based on three theoretical frameworks: (a) the impact of IR on WACC, (b) the association between IR and COE, and (c) the relation between IR and COD. We complement these hypotheses by postulating a possible moderating impact of IR on the link between ESG scores and WACC, COE, and COD:

(a)     Hypotheses regarding the relationship between IR and WACC

**H1a.** *"IR has a negative relationship with WACC".*

**H1b.** *"IR has a moderating impact on the ESG-WACC relationship".*

(b)     Hypotheses regarding the relationship between IR and COE

**H2a.** *"IR has a negative relationship with COE".*

**H2b.** *"IR has a moderating impact on the ESG-COE relationship".*

(c)     Hypotheses regarding the relationship between IR and COD.

**H3a.** *"IR has a negative relationship with COD".*

**H3b.** *"IR has a moderating impact on the ESG-COD relationship".*

To test our hypotheses, we employ a two-stage methodology. In the first stage, we conduct a comparative analysis between companies that prepare IR and those that do not, focusing on their ESG scores and cost of capital. We calculate the absolute difference in mean values between these two groups of firms. In the second stage, we employ panel data analysis to examine the direct impact of IR on the cost of capital.

Our baseline empirical model equation, which is specified to test for $H_{1a}$ and $H_{1b}$ is as follows:

$$\text{WACC}_{it} = \alpha + \beta_1 \text{IR}_{it} + \beta_2 \text{ESG}_{it} + \beta_3 \text{ESGxIR}_{it} + \beta_4 \text{CONTROL}_{it} + \varepsilon_{it} \qquad (1)$$

where WACC is the weighted average cost of capital, IR is a dummy variable of 1 when IR is prepared, ESG is the ESG score, and CONTROL is a vector of control variables including firm-specific and macroeconomic variables. The interaction term ESGxIR$_{it}$ is introduced in the model specification to capture a possible moderating effect of IR on the relationship between ESG and WACC. In other words, our model measures the direct impact of IR on WACC by $\beta_1$ and the indirect impact by $\beta_3$. To avoid potentially problematic high multicollinearity with the interaction term and to improve the interpretation of the results, the variables entering the interaction were mean-centered (Iacobucci et al. 2017).

We re-run Equation (1) by replacing WACC with COE to test for H2a and H2b and with COD to test for H3a and H3b. We also replace ESG with ENV, SOC, and GOV pillars to provide more information on the relationship between IR and the cost of capital.

Our panel dataset is characterized by being both unbalanced and short. It is unbalanced due to the absence of data for a few companies throughout the sample period. Additionally, it is short because although we have a large number of firms (59) in our sample (large N), we only have a limited number of time periods (6) (small T). To address these characteristics, we apply several techniques in our analysis. Firstly, we incorporate time effects and cluster the data to account for serial correlation. Additionally, we estimate both fixed and random effects models to capture the effects of individual firms. To determine the appropriate model specification, we employ the Hausman specification test (Hausman 1978), which compares the fixed effects and random effects models under the null hypothesis that individual effects are uncorrelated with any regressor in the model.

To ensure the robustness of our findings, we employ the two-step system-generalized method of moments (GMM) estimation method to estimate the parameters of the model. This method effectively addresses unobservable individual effects through first-order differencing, thereby minimizing their impact on the estimation. Additionally, it incorporates lagged instrumental variables to control for the potential association between the difference in the dependent variable and the error term. It is particularly useful for unbalanced panel data with large cross-sectional units (N) and small time (T) (Arellano and Bond 1991; Roodman 2009), such as in our case. Moreover, by employing the lags of the endogenous regressors as internal instruments, it mitigates potential endogeneity issues that may occur. In dynamic panel data models, the inclusion of lags of the dependent variable and the use of instruments are common methods to address endogeneity. By combining these approaches, the two-step system-GMM estimation method enables us to obtain consistent and efficient estimates (Roodman 2009).

In order to evaluate the soundness of the instruments used in our analysis and confirm the absence of second-order serial correlation, we conduct two statistical tests. Firstly, we employ the Hansen test of over-identification restrictions, which assesses whether the instrumental variables are valid and adequately instrument the endogenous regressors. This test helps ensure that the instrumental variables used in our model are exogenous and not correlated with the error term. Secondly, we conduct the Arellano and Bond's AR(2) test to examine the presence of second-order serial correlation in the differenced errors. This test is important for confirming the absence of autocorrelation and verifying the validity of the differencing technique in addressing the endogeneity issue.

## 3. Results

### 3.1. Univariate Test Results

Table 4 demonstrates the differences in means of the group of firms preparing IR compared to their non-preparing counterparts in terms of ESG and pillar scores.

As depicted in Table 4, the ESG scores of companies that prepare IR are significantly higher compared to those of companies that do not. This finding is consistent with previous studies by Lai et al. (2016) and Conway (2019). The significant differences in ESG scores between IR preparers and non-preparers are observed across various ESG pillars, particularly in the ENV and SOC pillars. Notably, the disparities in SOC pillar scores are particularly pronounced, indicating that IR preparers demonstrate greater success in achieving high

SOC pillar scores, as highlighted in Table 3. On the other hand, although the GOV pillar score is higher for IR preparers, the mean difference is not statistically significant.

**Table 4.** Mean Difference Test for the ESG/Pillar Scores of IR Preparers and Non-Preparers.

| Variable | Obs. | Mean | Std. Err. | *t*-Test | *p*-Value |
|---|---|---|---|---|---|
| $ESG_{NIR}$ | 200 | 58.476 | 1.082 | | |
| $ESG_{IR}$ | 23 | 66.764 | 2.626 | | |
| Diff. | | −8.288 | 2.841 | −2.918 *** | 0.000 |
| $ENV_{NIR}$ | 200 | 58.425 | 1.501 | | |
| $ENV_{IR}$ | 23 | 67.863 | 4.295 | | |
| Diff. | | −9.438 | 4.550 | −2.074 ** | 0.024 |
| $SOC_{NIR}$ | 200 | 64.019 | 1.396 | | |
| $SOC_{IR}$ | 23 | 75.920 | 3.053 | | |
| Diff. | | −11.901 | 3.357 | −3.545 *** | 0.000 |
| $GOV_{NIR}$ | 200 | 52.123 | 1.410 | | |
| $GOV_{IR}$ | 23 | 54.831 | 3.531 | | |
| Diff. | | −2.708 | 3.802 | −0.712 | 0.241 |

Note: This table shows the mean-differences between IR preparers and non-preparers in terms of their ESG and ENV, SOC, and GOV pillar scores. Subscripts NIR and IR stand for non-preparers and preparers of IR. Obs. defines the number of firm-year observations. *** and ** denote 1% and 5% significance levels, respectively.

However, it is important to consider that the higher ESG scores among IR preparers may not solely be attributed to the practice of IR. An alternative perspective suggests that companies may have already made significant progress in improving their ESG scores before implementing IR. Therefore, the observed increase in ESG scores may be influenced by factors other than IR alone.

Similarly, we compare the WACC, COE, and COD of IR preparers and non-preparers, and present the results in Table 5.

**Table 5.** Mean Difference Test for the WACC, COE, and COD of IR Preparers and Non-Preparers.

| Variable | Obs. | Mean | Std. Err. | *t*-Test | *p*-Value |
|---|---|---|---|---|---|
| $WACC_{NIR}$ | 200 | 0.088 | 0.002 | | |
| $WACC_{IR}$ | 23 | 0.078 | 0.004 | | |
| Diff. | | 0.009 | 0.004 | 2.292 ** | 0.014 |
| $COE_{NIR}$ | 200 | 0.138 | 0.002 | | |
| $COE_{IR}$ | 23 | 0.156 | 0.005 | | |
| Diff. | | −0.018 | 0.005 | −3.345 *** | 0.001 |
| $COD_{NIR}$ | 200 | 0.031 | 0.001 | | |
| $COD_{IR}$ | 23 | 0.037 | 0.004 | | |
| Diff. | | −0.006 | 0.004 | −1.568 * | 0.065 |

Note: This table shows the mean-differences between IR preparers and non-preparers in terms of their weighted average cost of capital (WACC), cost of equity (COE), and cost of debt (COD). Subscripts NIR and IR stand for non-preparers and preparers of IR. Obs. defines the number of firm-year observations. ***, **, and * denote 1%, 5%, and 10% significance levels, respectively.

According to Table 5, WACC for companies that prepare IR is significantly lower compared to companies that do not. These initial findings align with previous studies by (Zhou et al. 2017; García-Sánchez and Ligia 2017; Vena et al. 2020; Maama and Marimuthu 2021), suggesting that IR may play a role in reducing the cost of capital. However, it is worth noting that both COE and COD for IR preparers are significantly higher compared to their non-preparer counterparts. These results may appear contradictory to the WACC findings, but can be attributed to differences in the weighting of equity and debt in the firms' capital structure policies, which ultimately impact the overall WACC figures.

It is important to mention that these results are based on firm-year observations, meaning that a company may appear multiple times in the dataset during the sample period.

To account for this, we conducted mean-difference tests for the same set of companies that transitioned from non-preparers to preparers of IR, averaging their variables of interest. The results, not reported here for brevity, demonstrate similar qualitative patterns and are available upon request.

### 3.2. Panel Data Test Results

In our panel data analysis, we systematically present the results by following a step-wise approach. In Step 1, we investigate the direct impact of the main independent variable, IR, on WACC, COE, and COD. Subsequently, in Step 2, we introduce the ESG/pillar scores as additional variables. To further examine the potential moderating role of IR, Step 3 incorporates interaction variables. Finally, in Step 4, we present the comprehensive results of the full model encompassing all relevant variables and interactions.

Table 6 focuses on the relationship between IR and WACC, COE, and COD based on the overall ESG scores. We observe the direct impact of IR, finding a significant positive relationship with WACC (Panel A) and COD (Panel C). However, the relationship with COE (Panel B) is generally positive but statistically insignificant. Interestingly, the interaction variable $ESGxIR_{it}$, demonstrates a significant negative coefficient. This indicates that IR has the potential to moderate the significant relationship between ESG scores and WACC as well as COD. Specifically, in the case of WACC, the moderating role of IR reverses the impact of ESG on WACC. This suggests that if a company prepares IR, an increase in its ESG scores is likely to be associated with a lower WACC compared to non-preparers. However, in the case of COD, the moderating effect of IR is relatively weak. These findings highlight the potential of IR to influence the relationship between ESG scores and WACC or COD, showcasing its role as a moderator.

As for the control variables, the price-to-book ratio has a significant negative impact on COD. This suggests that firms with high growth potential may enjoy lower levels of borrowing costs because they are welcomed by creditors (Feng et al. 2021). Total debt is significantly related to WACC and COD. On the one hand, more leverage decreases the cost of capital, probably because of the tax shield property of debt. This is justifiable because the level of COD is very low compared to COE (see Table 3), and thus, WACC, which is a linear combination of COE and COD, is negatively related to leverage pointing toward tax shield benefits at lower levels of debt despite the risks increase with the share of debt (Singh et al. 2005; Evdokimova and Kuzubov 2021). On the other hand, leverage is associated with an increase in the COD, which is not surprising. CDS affects COE positively (Hong and Wang 2021), while it has a negative impact on COD. The positive relationship between CDS and COE can be derived from the Capital Asset Pricing Model. According to the model, the risk-free rate and eventually the equity risk premium should increase when CDS increases. The CDS-induced increase in credit risk should also translate into higher COD (Narayanan and Uzmanoglu 2018), but the prevailing unorthodox position in Turkey that lower interest rates could be the reason for the negative relationship between CDS and COD.

Next, Table 7a demonstrates the relationships considering each of the ESG pillars. In Panel A of Table 7a, we find that IR is positively associated with WACC. In addition, since the negative coefficient of the interaction variable, $ENVxIR_{it}$, is not significant, IR does not appear to play a moderating role in the relationship between ENV and WACC. While the results in regards to the SOC pillar in Panel B of Table 7a are quite similar, Panel C of Table 7a shows that the interaction variable, $GOVxIR_{it}$, has a significant negative impact on the GOV pillar-WACC nexus. Although IR has no moderating role due to the insignificant main effect of GOV on WACC, this finding implies that WACC can be lowered by increasing GOV pillar scores only for IR preparers.

**Table 6.** Impact of IR on WACC, COE, and COD (Overall ESG Scores).

| Panel A: IR-WACC Relationship with Respect to ESG | | | | | Panel B: IR-COE Relationship with Respect to ESG | | | | | Panel C: IR-COD Relationship with Respect to ESG | | | | |
|---|---|---|---|---|---|---|---|---|---|---|---|---|---|---|
| **WACC** | **Step 1** | **Step 2** | **Step 3** | **Step 4** | **COE** | **Step 1** | **Step 2** | **Step 3** | **Step 4** | **COD** | **Step 1** | **Step 2** | **Step 3** | **Step 4** |
| | **Coef. (std.err.)** | **Coef. (std.err.)** | **Coef. (std.err.)** | **Coef. (std.err.)** | | **Coef. (std.err.)** | **Coef. (std.err.)** | **Coef. (std.err.)** | **Coef. (std.err.)** | | **Coef. (std.err.)** | **Coef. (std.err.)** | **Coef. (std.err.)** | **Coef. (std.err.)** |
| IR | 0.004 (0.003) | 0.004 * (0.002) | 0.007 ** (0.003) | 0.004 (0.003) | IR | −0.000 (0.003) | 0.000 (0.003) | 0.002 (0.005) | 0.002 (0.005) | IR | 0.006 ** (0.003) | 0.006 ** (0.003) | 0.010 *** (0.004) | 0.011 *** (0.003) |
| ESG | | 0.007 ** (0.003) | 0.007 * (0.003) | 0.007 * (0.004) | ESG | | −0.001 (0.007) | −0.001 (0.007) | −0.001 (0.007) | ESG | | | −0.009 ** (0.004) | −0.009 ** (0.004) | −0.009 ** (0.004) |
| ESGxIR | | | −0.015 * (0.008) | −0.023 ** (0.009) | ESGxIR | | | −0.013 (0.021) | −0.015 (0.021) | ESGxIR | | | | −0.025 * (0.015) | -0.021 (0.014) |
| PB | | | | 0.001 (0.001) | PB | | | | −0.001 (0.000) | PB | | | | | −0.001 *** (0.000) |
| LEV | | | | −0.045 ** (0.022) | LEV | | | | −0.010 (0.017) | LEV | | | | | 0.034 ** (0.016) |
| TA | | | | −0.011 (0.010) | TA | | | | −0.003 (0.005) | TA | | | | | −0.001 (0.006) |
| CDS | | | | −0.006 (0.006) | CDS | | | | 0.034 *** (0.005) | CDS | | | | | −0.011 * (0.006) |
| Year fixed-effects | Yes | Yes | Yes | Yes | | Yes | Yes | Yes | Yes | | Yes | Yes | Yes | Yes |
| Hausman Test | 11.520 ** | 12.970 ** | 13.570 * | 16.810 * | | 5.910 | 5.980 | 8.460 | 20.690 ** | | 6.060 | 9.720 | 14.350 ** | 13.390 |
| # of obs. | 223 | 223 | 223 | 221 | | 223 | 223 | 223 | 221 | | 223 | 223 | 223 | 221 |
| R sq. F | 0.194 43.96 *** | 0.171 37.32 *** | 0.170 33.56 *** | 0.511 33.68 *** | R sq. F | 0.542 135.25 *** | 0.541 153.67 *** | 0.537 128.87 *** | 0.456 101.55 *** | R sq. F | 0.329 31.04 *** | 0.287 27.02 *** | 0.264 22.54 *** | 0.271 24.64 *** |

Note: This table shows the panel data analysis results on the relationship between IR and weighted average cost of capital (WACC), cost of equity (COE), and cost of debt (COD), when overall ESG scores are considered. ***, **, and * denote 1%, 5%, and 10% significance levels, respectively.

We follow the same approach in presenting the results for the relationship between IR and COE as well as IR and COD in the context of the ESG pillars in Tables 7b and 7c, respectively. Table 7b suggests that none of the variables of interest are statistically significant. However, the results are interesting in that the coefficient of IR is negative, reflecting the likelihood of a decrease in COE as IR practices improve. We report significant findings in Table 7c. First, we find a significant positive relationship between IR and COD in each case in Panel A, Panel B, and Panel C of Table 7c. Accordingly, IR preparers have higher COD. We observe the moderating impact of IR only on the relationship between SOC and COD. We also infer that COD can be alleviated by increasing the GOV pillar scores for IR preparers, which is a similar conclusion when considering the WACC-GOV relationship in Panel C of Table 6.

The negative relationship between the price-to-book ratio and COD, which we have reported earlier, can also be traced in Table 7c. Again, total debt has a significant negative influence on WACC (Table 7a), while its effect on COD is broadly positive (Table 7c). Moreover, the effects of CDS on COE (Table 7b) and COD (Table 7c) are consistent with our previous findings.

### 3.3. Robustness Check

To estimate Equation (1), we employ a two-step system-GMM estimation method, which accounts for the dynamic nature of the data by using the lagged values of the dependent variables (WACC, COE, and COD) as instruments (García-Sánchez and Ligia 2017; Muttakin et al. 2020; Maama and Marimuthu 2021). This approach combines both the GMM estimator and instrumental variable (IV) estimator to address the issue of endogeneity. In the GMM estimation, we use lags (3 3) of the endogenous firm-level variables as instruments to ensure their exogeneity and prevent their correlation with the error term in the regression equation. Additionally, we employ IR, ESG/pillar, and interaction variables as exogenous instruments to further control for endogeneity. To correct for heteroscedasticity, we use robust standard errors. We also include year variables to account for time fixed effects, which capture systematic differences in the WACC, COE, and COD variables specific to each year but not specific to individual firms. The number of instruments used (28) is less than the number of groups (47). We conduct misspecification tests, including the AR(2) test for second-order serial correlation and the Hansen test, for each model. The results of these tests indicate no evidence of serial correlation and instrument validity, confirming that the models are appropriately specified and produce reliable estimates. Thus, the instrumental variables employed in the GMM estimation are uncorrelated with the error term and correlated with the endogenous variables, indicating that they effectively address concerns related to endogeneity.

We only report the outcomes of Step 4 in Table 8. We first find that the lagged cost of financing (WACC, COE, and COD) has a positive and significant impact on the prevailing cost of financing in all models, which is in line with Maama and Marimuthu (2021). The results also suggest that IR has a consistent and positive impact on firms' cost of financing, which supports our previous findings. The interaction variables have negative coefficients implying that IR has the potential to moderate the relationship between ESG scores and cost of capital. However, due to the statistically insignificant effects of ESG/pillar scores (except for GOV in Panel A), it is more reasonable to conclude that cost of capital can be lowered by increasing ESG/pillar scores only for IR preparers.

**Table 7.** (**a**) Impact of IR on WACC (ESG Pillar Scores). (**b**) Impact of IR on COE (ESG Pillar Scores). (**c**) Impact of IR on COD (ESG Pillar Scores).

(**a**)

| | Panel A: IR-WACC Relationship with Respect to ENV | | | | | Panel B: IR-WACC Relationship with Respect to SOC | | | | | Panel C: IR-WACC Relationship with Respect to GOV | | | |
|---|---|---|---|---|---|---|---|---|---|---|---|---|---|---|
| WACC | Step 1 | Step 2 | Step 3 | Step 4 | | Step 1 | Step 2 | Step 3 | Step 4 | | Step 1 | Step 2 | Step 3 | Step 4 |
| | Coef. (std.err.) | Coef. (std.err.) | Coef. (std.err.) | Coef. (std.err.) | | Coef. (std.err.) | Coef. (std.err.) | Coef. (std.err.) | Coef. (std.err.) | | Coef. (std.err.) | Coef. (std.err.) | Coef. (std.err.) | Coef. (std.err.) |
| IR | 0.004 (0.003) | 0.005 * (0.003) | 0.006 ** (0.003) | 0.003 (0.003) | IR | 0.004 (0.003) | 0.004 * (0.003) | 0.005 ** (0.002) | 0.003 (0.003) | IR | 0.004 (0.003) | 0.004 (0.003) | 0.004 * (0.002) | 0.000 (0.000) |
| ENV | | 0.002 ** (0.001) | 0.002 ** (0.001) | 0.003 *** (0.001) | SOC | | | 0.005 * (0.003) | 0.005 * (0.003) | GOV | | | 0.002 (0.003) | 0.002 (0.003) |
| ENVxIR | | | −0.005 (0.004) | −0.008 (0.006) | SOCxIR | | | | −0.004 (0.005) | GOVxIR | | | −0.010 *** (0.003) | −0.012 *** (0.004) |
| PB | | | | 0.001 (0.001) | PB | | | | 0.005 (0.003) | PB | | | | 0.001 (0.001) |
| LEV | | | | −0.046 ** (0.022) | LEV | | | | −0.009 (0.008) | LEV | | | | −0.043 * (0.022) |
| TA | | | | −0.012 (0.010) | TA | | | | 0.001 (0.001) | TA | | | | −0.012 (0.010) |
| CDS | | | | −0.005 (0.005) | CDS | | | | −0.04 6 ** (0.023) | CDS | | | | −0.003 (0.005) |
| | | | | | | | | | −0.011 (0.010) | | | | | |
| | | | | | | | | | −0.006 (0.006) | | | | | |
| Year fixed-effects | Yes | Yes | Yes | Yes | | Yes | Yes | Yes | Yes | | Yes | Yes | Yes | Yes |
| Hausman Test | 11.520 ** | 14.870 ** | 14.680 * | 15.820 | | 11.520 ** | 12.840 ** | 12.790 * | 13.840 | | 11.520 ** | 13.100 ** | 15.930 ** | 25.220 *** |
| # of obs. | 223 | 222 | 222 | 220 | | 223 | 223 | 223 | 221 | | 223 | 223 | 223 | 221 |
| R sq. | 0.194 | 0.176 | 0.177 | 0.509 | R sq. | 0.194 | 0.165 | 0.165 | 0.519 | R sq. | 0.194 | 0.203 | 0.204 | 0.496 |
| F | 43.96 *** | 39.64 *** | 43.41 *** | 34.88 *** | F | 43.96 *** | 36.44 *** | 32.04 *** | 34.93 *** | F | 43.96 *** | 37.80 *** | 40.26 *** | 37.51 *** |

**Table 7.** *Cont.*

| | (b) | | | | | | | | | | | | | |
|---|---|---|---|---|---|---|---|---|---|---|---|---|---|---|
| | **Panel A: IR-COE Relationship with Respect to ENV** | | | | | **Panel B: IR-COE Relationship with Respect to SOC** | | | | | **Panel C: IR-COE Relationship with Respect to GOV** | | | |
| **COE** | **Step 1** | **Step 2** | **Step 3** | **Step 4** | | **Step 1** | **Step 2** | **Step 3** | **Step 4** | | **Step 1** | **Step 2** | **Step 3** | **Step 4** |
| | **Coef. (std.err.)** | **Coef. (std.err.)** | **Coef. (std.err.)** | **Coef. (std.err.)** | | **Coef. (std.err.)** | **Coef. (std.err.)** | **Coef. (std.err.)** | **Coef. (std.err.)** | | **Coef. (std.err.)** | **Coef. (std.err.)** | **Coef. (std.err.)** | **Coef. (std.err.)** |
| IR | −0.000 (0.003) | −0.001 (0.003) | −0.001 (0.005) | −0.001 (0.005) | IR | −0.000 (0.003) | −0.000 (0.003) | 0.003 (0.005) | 0.003 (0.005) | IR | −0.000 (0.003) | −0.000 (0.003) | −0.000 (0.003) | −0.001 (0.004) |
| ENV | | −0.001 (0.000) | −0.001 (0.000) | −0.001 (0.002) | SOC | | −0.003 (0.005) | −0.003 (0.005) | −0.003 (0.005) | GOV | | | 0.002 (0.003) | 0.002 (0.003) |
| ENVxIR | | | 0.001 (0.014) | 0.000 (0.014) | SOCxIR | | | −0.014 (0.016) | -0.017 (0.017) | GOVxIR | | | −0.002 (0.010) | −0.002 (0.010) |
| PB | | | | −0.001 (0.000) | PB | | | | −0.001 (0.000) | PB | | | | −0.001 (0.000) |
| LEV | | | | −0.009 (0.017) | LEV | | | | −0.012 (0.017) | LEV | | | | −0.007 (0.017) |
| TA | | | | −0.002 (0.005) | TA | | | | −0.002 (0.006) | TA | | | | −0.002 (0.006) |
| CDS | | | | 0.034 *** (0.005) | CDS | | | | 0.035 *** (0.005) | CDS | | | | 0.032 *** (0.005) |
| Year fixed-effects | Yes | Yes | Yes | Yes | | Yes | Yes | Yes | Yes | | Yes | Yes | Yes | Yes |
| Hausman Test | 5.910 | 8.570 | 12.370 | 21.510 ** | | 5.910 | 6.130 | 12.400 * | 29.230 *** | | 5.910 | 5.950 | 6.380 | 20.150 ** |
| # of obs. | 223 | 222 | 222 | 220 | | 223 | 223 | 223 | 221 | | 223 | 223 | 223 | 221 |
| R sq. F | 0.542 135.25 *** | 0.536 148.06 *** | 0.537 131.03 *** | 0.481 103.34 *** | R sq. F | 0.542 135.25 *** | 0.534 159.02 *** | 0.528 131.47 *** | 0.470 105.02 *** | R sq. F | 0.542 135.25 *** | 0.538 182.18 *** | 0.538 170.15 *** | 0.467 142.49 *** |

**Table 7.** *Cont.*

**(c)**

| | Panel A: IR-COD Relationship with Respect to ENV | | | | | Panel B: IR-COD Relationship with Respect to SOC | | | | | Panel C: IR-COD Relationship with Respect to GOV | | | |
|---|---|---|---|---|---|---|---|---|---|---|---|---|---|---|
| COD | Step 1 | Step 2 | Step 3 | Step 4 | | Step 1 | Step 2 | Step 3 | Step 4 | | Step 1 | Step 2 | Step 3 | Step 4 |
| | Coef. (std.err.) | Coef. (std.err.) | Coef. (std.err.) | Coef. (std.err.) | | Coef. (std.err.) | Coef. (std.err.) | Coef. (std.err.) | Coef. (std.err.) | | Coef. (std.err.) | Coef. (std.err.) | Coef. (std.err.) | Coef. (std.err.) |
| IR | 0.006 ** (0.003) | 0.006 * (0.002) | 0.007 * (0.004) | 0.009 ** (0.004) | IR | 0.006 ** (0.003) | 0.006 ** (0.003) | 0.010 *** (0.003) | 0.011 *** (0.003) | IR | 0.006 ** (0.003) | 0.006 ** (0.003) | 0.006 ** (0.003) | 0.008 *** (0.003) |
| ENV | | −0.001 (0.003) | −0.001 (0.002) | −0.001 (0.002) | SOC | | −0.009 ** (0.004) | −0.009 ** (0.004) | −0.009 ** (0.004) | GOV | | −0.000 (0.003) | 0.001 (0.004) | 0.002 (0.004) |
| ENVxIR | | | −0.006 (0.011) | −0.004 (0.010) | SOCxIR | | | −0.019 (0.011) | −0.013 * (0.009) | GOVxIR | | | −0.010 ** (0.005) | −0.011 ** (0.005) |
| PB | | | | −0.001 *** (0.000) | PB | | | | −0.001 *** (0.000) | PB | | | | −0.001 *** (0.000) |
| LEV | | | | −0.038 ** (0.017) | LEV | | | | 0.035 ** (0.016) | LEV | | | | 0.040 ** (0.016) |
| TA | | | | −0.000 (0.006) | TA | | | | −0.000 (0.006) | TA | | | | −0.002 (0.006) |
| CDS | | | | −0.014 ** (0.006) | CDS | | | | −0.010 * (0.006) | CDS | | | | −0.015 *** (0.005) |
| Year fixed-effects | Yes | Yes | Yes | Yes | | Yes | Yes | Yes | Yes | | Yes | Yes | Yes | Yes |
| Hausman Test | 6.060 | 7.110 | 12.280 | 11.370 | | 6.060 | 10.950 * | 12.260 * | 12.800 | | 6.060 | 5.950 | 6.380 | 20.150 ** |
| # of obs. | 223 | 222 | 222 | 220 | | 223 | 223 | 223 | 221 | | 223 | 223 | 223 | 221 |
| R sq. | 0.329 | 0.321 | 0.306 | 0.322 | R sq. | 0.329 | 0.259 | 0.249 | 0.284 | R sq. | 0.329 | 0.330 | 0.321 | 0.224 |
| F | 31.04 *** | 27.17 *** | 23.65 *** | 28.80 *** | F | 31.04 *** | 25.45 *** | 22.18 *** | 24.15 *** | F | 31.04 *** | 26.47 *** | 22.36 *** | 25.89 *** |

Note: This table **a** shows the panel data analysis results on the relationship between IR and weighted average cost of capital (WACC), when ENV, SOC, and GOV pillar scores are considered. ***, **, and * denote 1%, 5%, and 10% significance levels, respectively. This table **b** shows the panel data analysis results on the relationship between IR and cost of equity (COE), when ENV, SOC, and GOV pillar scores are considered. ***, **, and * denote 1%, 5%, and 10% significance levels, respectively. This table **c** shows the panel data analysis results on the relationship between IR and cost of debt (COD), when ENV, SOC, and GOV pillar scores are considered. ***, **, and * denote 1%, 5%, and 10% significance levels, respectively.

**Table 8.** Impact of IR on WACC, COE, and COD (ESG/Pillar Scores).

| | Panel A: IR-WACC Relationship | | | | | Panel B: IR-COE Relationship | | | | | Panel C: IR-COD Relationship | | | |
|---|---|---|---|---|---|---|---|---|---|---|---|---|---|---|
| **WACC** | **ESG** | **ENV** | **SOC** | **GOV** | **COE** | **ESG** | **ENV** | **SOC** | **GOV** | **COD** | **ESG** | **ENV** | **SOC** | **GOV** |
| | **Coef. (std.err.)** | **Coef. (std.err.)** | **Coef. (std.err.)** | **Coef. (std.err.)** | | **Coef. (std.err.)** | **Coef. (std.err.)** | **Coef. (std.err.)** | **Coef. (std.err.)** | | **Coef. (std.err.)** | **Coef. (std.err.)** | **Coef. (std.err.)** | **Coef. (std.err.)** |
| WACC(-1) | 0.527 ** (0.223) | 0.528 ** (0.202) | 0.405 ** (0.199) | 0.540 *** (0.163) | COE(-1) | 0.915 *** (0.129) | 0.886 *** (0.146) | 0.910 *** (0.129) | 0.905 *** (0.149) | COD(-1) | 0.832 *** (0.232) | 0.790 *** (0.211) | 0.921 *** (0.247) | 0.698 *** (0.232) |
| IR | 0.004 (0.005) | 0.004 (0.004) | 0.004 (0.003) | 0.003 (0.003) | IR | 0.000 (0.007) | −0.006 (0.012) | −0.000 (0.009) | −0.004 (0.007) | IR | 0.005 ** (0.003) | 0.004 (0.003) | 0.007 ** (0.003) | 0.002 (0.004) |
| ESG/Pillar | 0.004 (0.009) | 0.002 (0.002) | −0.006 (0.008) | 0.006 ** (0.003) | ESG/Pillar | 0.003 (0.005) | 0.002 (0.001) | 0.003 (0.005) | -0.000 (0.003) | ESG/Pillar | −0.002 (0.004) | -0.001 (0.001) | −0.003 (0.003) | −0.000 (0.003) |
| ESG/PillarxIR | −0.032 (0.019) | −0.008 (0.009) | −0.019 (0.012) | −0.016 * (0.008) | ESG/PillarxIR | −0.027 (0.030) | 0.010 (0.028) | −0.016 (0.025) | −0.007 (0.014) | ESG/PillarxIR | −0.035 ** (0.014) | −0.011 (0.007) | −0.030 *** (0.009) | −0.013 ** (0.006) |
| PB | 0.006 (0.005) | 0.005 (0.005) | 0.007 * (0.004) | 0.005 (0.003) | PB | 0.000 (0.003) | −0.000 (0.003) | 0.000 (0.003) | −0.001 (0.003) | PB | 0.000 (0.002) | −0.000 (0.002) | 0.000 (0.001) | −0.000 (0.002) |
| LEV | −0.055 * (0.028) | −0.054 * (0.029) | −0.065 *** (0.019) | −0.049 *** (0.018) | LEV | −0.004 (0.011) | −0.001 (0.010) | −0.005 (0.012) | −0.001 (0.010) | LEV | 0.005 (0.011) | 0.010 (0.011) | 0.002 (0.011) | 0.013 (0.011) |
| TA | −0.002 (0.002) | −0.003 (0.002) | −0.003 (0.002) | −0.003 ** (0.002) | TA | −0.000 (0.002) | −0.001 (0.002) | −0.000 (0.002) | −0.001 (0.003) | TA | 0.003 (0.002) | 0.002 (0.002) | 0.002 (0.002) | 0.002 (0.002) |
| CDS | −0.013 (0.019) | −0.015 (0.017) | - - | - - | CDS | - - | −0.050 *** (0.009) | - - | - - | CDS | - - | -0.004 (0.004) | - - | - - |
| Year fixed-effects | Yes | Yes | Yes | Yes | | Yes | Yes | Yes | Yes | | Yes | Yes | Yes | Yes |
| AR(2) *p*-value | 0.848 | 0.860 | 0.781 | 0.879 | | 0.170 | 0.186 | 0.192 | 0.184 | | 0.148 | 0.123 | 0.107 | 0.163 |
| Hansen *p*-value | 0.203 | 0.193 | 0.255 | 0.363 | | 0.294 | 0.314 | 0.210 | 0.097 | | 0.560 | 0.470 | 0.780 | 0.295 |
| # of obs. | 163 | 163 | 163 | 163 | | 163 | 163 | 163 | 163 | | 163 | 163 | 163 | 163 |
| # of grp./inst. | 47/28 | 47/28 | 47/28 | 47/28 | | 47/28 | 47/28 | 47/28 | 47/28 | | 47/28 | 47/28 | 47/28 | 47/28 |

Note: This table shows the two-step system-GMM estimation results on the relationship between IR and weighted average cost of capital (WACC), cost of equity (COE), and cost of debt (COD), when ESG/pillar scores are considered. ***, **, and * denote 1%, 5%, and 10% significance levels, respectively.

## 4. Discussion

The results we outline above are summarized along with the relevant hypotheses in Appendix B.

Consistent with these results, we reject H1a, H2a, and H3a because IR has no negative relationship with WACC, COE, and COD. While we reject H2b, we cannot reject $H_{1b}$ and $H_{3b}$ due to the fact that the interaction variable, $ESGxIR_{it}$, may affect the direction and the magnitude of the relationship between ESG and WACC as well as ESG and COD.

The positive impact of IR on WACC is not in line with our expectations, but it adds to our findings on the relationship between ESG/pillar scores and WACC. As our results suggest, WACC increases as ESG and its pillars are better scored, and similarly, WACC is higher for companies that prepare IR. Although this appears to contradict our literature-based hypothesis, it is reasonable to argue that ESG and IR practices may not yet be perceived positively in an emerging market because such practices are seen as so expensive that they require too much capital to pursue. Another possible explanation would be that IR can bring forth more complexity, making the decision-making process less efficient (Lodhia 2015) and harming the understanding of information presented (Reimsbach et al. 2018). On the other hand, our findings regarding the moderating effect of IR on the relationship between ESG and WACC do not corroborate these arguments, indicating that companies that care about sustainability can reap the benefits of IR by lowering their WACC. In other words, WACC reduction seems possible if "sustainable" companies also use IR to better communicate their value creating activities. Our results regarding the moderating role of IR are consistent with those of (Albitar et al. 2019), who showed that IR moderates the relationship between ESG disclosure and financial performance. (Barth et al. 2017; Mervelskemper and Streit 2017; Karwowski and Raulinajtys-Grzybek 2021; Rabaya and Saleh 2022) also support our findings on the reinforcing role of IR in the ESG context.

When examining the impact of ESG and IR on COE, we find that neither ESG nor IR has a significant influence on COE. This suggests that stock market investors may not place a significant emphasis on contemporary sustainability practices or may have reservations about their relevance. It is possible that investors do not perceive the information disclosed in IR as useful for making resource allocation decisions or achieving cost reductions. Previous research by (Steyn 2014; Lodhia 2015) supports the notion that investors may not fully appreciate the value of sustainability practices as disclosed in IR. The lack of significance in the relationship between COE, ESG, and IR also indicates that IR does not have a moderating effect on the relationship between COE and ESG or its individual pillars. However, it is important to note that both ESG and IR have the potential to play a mitigating role in relation to their negative coefficients. This suggests that as investors become more sophisticated and gain a deeper understanding of sustainability practices over time, the capital market may witness a decline in COE. While the current findings do not show a direct impact of ESG and IR on COE, it is plausible to anticipate that as the market evolves and investors become more knowledgeable about the value of sustainability practices, COE could be influenced by ESG factors and IR. Future research may shed more light on the dynamics between COE, ESG, and IR as investor attitudes and market perceptions evolve.

Regarding the relationship between IR and COD, we observe a different pattern. We find that high ESG scores translate into low borrowing costs, which is in line with (Luo et al. 2019; Eliwa et al. 2021; Raimo et al. 2022). This means that creditors favor companies with higher ESG scores when setting their lending rates. So, unlike capital market investors, creditors seem to be aware of and appreciate the benefits of sustainability practices. Interestingly, however, IR is positively related to COD, suggesting that IR preparers have higher COD, which is similar to our findings on WACC. IR, by itself, does not appear to lower borrowing costs, probably because they are viewed as subjective or opaque. Our inference is that lenders may not consider IR as a value enhancing or risk reducing tool for the firm as it has been documented in the sustainability literature (Goss and Roberts 2011). This can also be associated with a specific characteristic of the emerging Turkish market. Indeed, Kılıç and Kuzey (2018) argue that IR in Turkey

ignore company-specific risks; dismiss negative information; lack a strategic focus; and exclude forward-looking information. However, the social pillar seems to be of particular importance because of the moderating effect of IR on the SOC-COD relationship. In this way, socially sensitive IR preparers may still take the advantage of lower costs in the debt market.

Finally, it is worth noting the consistently negative coefficients of the interaction variables in our models. Even though only the results regarding the governance pillar reveal significance in the sense that GOV score lowers WACC and COD only for IR preparers, these negative signs suggest that IR has a potential to lower funding costs among "sustainable" companies. These findings imply that the combination of IR and strong ESG performance can lead to a decrease in the cost of capital for companies that embrace sustainable practices. By effectively communicating their sustainability efforts through IR, these firms may gain credibility and trust from investors and creditors, resulting in lower funding costs. While the significance is evident in relation to the GOV pillar, the negative signs of the interaction coefficients suggest that similar effects could potentially be observed for other ESG pillars as well.

Overall, these results highlight the potential benefits of IR and sustainable practices in reducing funding costs and enhancing financial performance for companies committed to sustainability.

## 5. Conclusions

In the contemporary investment environment, investors exhibit a growing interest in comprehending not only the financial risks, but also the non-financial risks associated with companies, as well as how these risks are effectively managed and how value is created over short-, medium- and long-term horizons. As a consequence, there is an increasing demand for enhanced transparency in corporate disclosures, encompassing corporate strategy, business model and ESG performance. This augmented level of transparency aims to alleviate uncertainties that may impede informed investment decision-making. In this context, IR emerges as a comprehensive reporting approach that portrays a company's performance in a holistic manner, offering a structured framework that furnishes investors with the requisite information to ascertain the true value of the company.

It is widely recognized in the literature that IR is a means for companies to gain competitive advantage through cost reduction, operational efficiency, brand value enhancement, and innovation. Moreover, the adoption of IR facilitates greater transparency and the provision of high-quality reporting, which in turn cultivates investor confidence in the company. This heightened confidence not only strengthens the company's reputation but also facilitates its access to funds, thereby streamlining the capital-raising process.

In this study, we investigate the relationship between IR and WACC, COE, and COD for a sample of 59 companies included in the XUSRD during 2015–2020. We also examine the moderating role of IR on the relationship between ESG scores and the cost of capital.

Our results provide theoretical insights by revealing that IR is positively associated with WACC and COD, while it has no significant impact on COE. This suggests that investors in Turkish emerging capital markets may not fully recognize the benefits of IR in terms of reducing financing costs. Thus, we conclude that investors have not yet started to give sufficient credit for IR in Turkey. However, the moderating effect of IR on the ESG-WACC and ESG-COD relationships indicates that WACC and COD can be lowered if "sustainable" companies also use IR to better communicate their value creating activities. These findings add to the growing body of knowledge on the evolving understanding of integrated reporting, the determinants of the cost of financing, investor decision-making processes, and the contextualization of IR in emerging capital markets.

These results have important managerial and practical implications as well. Managers should be aware that IR alone may not automatically lead to lower financing costs. It is crucial for companies to identify and address the factors that may counteract the potential benefits of IR, such as the lack of transparency, objectivity, and holistic reporting. By

improving the quality and effectiveness of IR, companies can enhance their communication with investors and creditors, potentially leading to reduced WACC and COD. Managers should also consider integrating sustainable practices and ESG disclosures into their IR processes, as this study suggests that the moderating effect of IR on the relationship between ESG scores and the cost of capital can contribute to lower financing costs. Investors and creditors should recognize the significance of integrated reporting as a valuable tool for assessing the true value and sustainability performance of companies. By developing their understanding of IR and integrated thinking, investors can make more informed decisions and allocate their resources effectively. Regulators and policymakers should prioritize the promotion and awareness of IR through the development of guidelines and frameworks that encourage market participants to adopt comprehensive and transparent reporting practices.

Our study is not without limitations. The unavailability of data poses a challenge, particularly in comparing preparers and non-preparers of IR in a larger sample of firms. For instance, we cannot compare preparers and non-preparers of IR in a larger sample of firms, since there is no IR preparer firm outside of XUSRD. Furthermore, our sample period is limited to 2015–2020, and the study focuses on Turkish emerging capital markets. Future research should address these limitations by exploring the relationship between IR and financial performance across different industries, incorporating a broader sample, and considering a longer time span. In addition, analyzing the content of IR to reveal their quality from a language perspective warrants another research area.

**Author Contributions:** Conceptualization, B.P. and L.R., methodology, B.P., software, B.P., formal analysis, B.P., investigation, B.P. and L.R., resources, B.P. and L.R., data curation, B.P., writing—original draft preparation, B.P. and L.R., visualization, B.P. and L.R., validation, B.P. and L.R. All authors have read and agreed to the published version of the manuscript.

**Funding:** This research received no external funding.

**Institutional Review Board Statement:** Not applicable.

**Informed Consent Statement:** Not applicable.

**Data Availability Statement:** The data presented in this study are available on request from the corresponding author. The data are not publicly available due to the fact that they are obtained from the Refinitiv database.

**Conflicts of Interest:** The authors declare no conflict of interest.

## Appendix A

*Institutional Framework for Turkish IR System*

IR has become a priority issue in Turkey since 2011, when the Corporate Governance Association of Turkey and the Business Council for Sustainable Development Turkey launched a working group to raise awareness of IR in Turkish capital markets. The working group prepared a project called "New Era in Corporate Reporting: IR" in 2013 and a formal guidance on IR in Turkey was published in 2015 (Aras and Sarioglu 2015). In 2016, the IR Turkey Network (ERTA) was founded and the first IR was published by Arguden Governance Academy, a non-governmental organization, later that year. Borsa Istanbul (BIST) and the International IR Council (IIRC) signed a cooperation agreement in 2017 to disseminate information on IR in Turkey. The following year, BIST became the first European stock exchange to publish an IR. Another achievement in 2018 was the protocol that made ERTA an official international partner of the IIRC. In collaboration with the Turkish Investor Relations Society, ERTA built a "IR & Investor Relations Platform" in 2020. The major goal of the platform is to assist companies in IR and improve communication between them and their investors. As a final step, ERTA and BIST published an "IR Guide for Companies" in April 2022 to support companies in publishing an IR.

On the other hand, sustainability reporting practices in Turkey date back to 2005. Since then, awareness of sustainability reports and the importance placed on them has gradually increased. BIST and the Capital Markets Board of Turkey (CMB) are credited with encouraging Turkish listed companies to take more stakes from global sustainable investment flows. Recently, BIST updated its 2014 Sustainability Guide for Companies, which provides companies with a roadmap for ESG issues, and the CMB issued a regulation requiring companies to disclose information in a "Sustainability Principles Compliance Framework" on whether or not sustainability principles are applied and, if not, to provide a reasoned explanation. Another key initiative to promote sustainability was the launch of the BIST Sustainability Index (XUSRD) in 2014. The constituents of the XUSRD are selected from a list of companies, which are traded on the markets of BIST and are shares of companies with high sustainability performance. In 2021, the calculation methodology was revised to reflect the ESG scores from the data company Refinitiv. Today, companies wishing to be included in the XUSRD must meet three criteria: (a) the overall ESG score should be at least 50; (b) each pillar score should be at least 40; and (c) at least 8 of the category scores should be 26 or more.

## Appendix B

**Table A1.** Summary of Findings.

| Notation | Result | Relationship |
| --- | --- | --- |
| H1a | Rejection | Significant (+) |
| H1b | No Rejection | Moderation (−) |
| H2a | Rejection | Insignificant (+,−) |
| H2b | Rejection | No Moderation (−) |
| H3a | Rejection | Significant (+) |
| H3b | No Rejection | Moderation (−) |

Note: This table summarizes the results of our analyses.

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
