# Peer review of "The Impact of Integrated Reporting on the Cost of Capital: Evidence from an Emerging Market"

_jrfm, doi:10.3390/jrfm16070311_

Round 1

Reviewer 1 Report

First of all, I would like to thank you for the possibility of reviewing this interesting paper that I have read with great interest.

The paper may have a clear interest associated to researchers from different scientific disciplines and, therefore, could have a notable repercussion in specialized scientific literature.

Method is well

Conclusions: pleas add theoretical, managerial, and practical implications, limitation and further research. Some parts are included but must be extended.

Overall, I believe that the ideas are well expressed, and the storyline is easily followed by the reader. However, in the course of reading the manuscript, I could identify some minor mistakes that should be dealt with more carefully by the authors.

Good luck!

Reviewer 2 Report

Dear authors, Your paper addresses an interesting topic in a novel context. You show an unexpected result: IR concerns increase the cost of capital. This at-odds result deserves better testing.

i. endogeneity. Small t large N, why not GMM estimation, or using lags of independent variables, or panel regression on instruments? In fact, you only test for the contemporaneous relation.

ii. Given your title and abstract, it seems that your first step should be a regression of the cost of capital on IR (not on ESG) and then enter the other tests (if relevant)

Round 2

Reviewer 2 Report

Dear authors, you improved your paper, as suggested. Please note that, as stated above, I am not qualified to assess the quality of English.